# The Role of WRAP53 in Cell Homeostasis and Carcinogenesis Onset

Renan Brito Gadelha [1,†], Caio Bezerra Machado [1,†], Flávia Melo Cunha de Pinho Pessoa [1], Laudreísa da Costa Pantoja [2,3], Igor Valentim Barreto [1], Rodrigo Monteiro Ribeiro [4], Manoel Odorico de Moraes Filho [1], Maria Elisabete Amaral de Moraes [1], André Salim Khayat [3] and Caroline Aquino Moreira-Nunes [1,3,5,*]

1  Pharmacogenetics Laboratory, Department of Medicine, Drug Research and Development Center (NPDM), Federal University of Ceará, Fortaleza 60430-275, CE, Brazil
2  Department of Pediatrics, Octávio Lobo Children's Hospital, Belém 60430-275, PA, Brazil
3  Department of Biological Sciences, Oncology Research Center, Federal University of Pará, Belém 66073-005, PA, Brazil
4  Department of Hematology, Fortaleza General Hospital (HGF), Fortaleza 60150-160, CE, Brazil
5  Northeast Biotechnology Network (RENORBIO), Itaperi Campus, Ceará State University, Fortaleza 60740-903, CE, Brazil
*  Correspondence: carolfam@gmail.com
†  These authors contributed equally to this work.

**Abstract:** The WD repeat containing antisense to TP53 (*WRAP53*) gene codifies an antisense transcript for tumor protein p53 (*TP53*), stabilization (WRAP53α), and a functional protein (WRAP53β, WDR79, or TCAB1). The WRAP53β protein functions as a scaffolding protein that is important for telomerase localization, telomere assembly, Cajal body integrity, and DNA double-strand break repair. WRAP53β is one of many proteins known for containing WD40 domains, which are responsible for mediating a variety of cell interactions. Currently, *WRAP53* overexpression is considered a biomarker for a diverse subset of cancer types, and in this study, we describe what is known about WRAP53β's multiple interactions in cell protein trafficking, Cajal body formation, and DNA double-strand break repair and its current perspectives as a biomarker for cancer.

**Keywords:** *WRAP53*; genomic instability; carcinogenesis; TP53; telomerase

## 1. Introduction

Cancers arise through a series of mutations or genetic alterations that give the cell the ability to override pro-apoptotic and anti-proliferative signals, allowing it to reach the hallmarks of replicative immortality, invasion, and metastasis [1–3].

Several molecular mechanisms work to make it possible for cells to acquire genetic changes, whether at the level of nucleotides or chromosomes. Tumorigenesis is seen as an imbalance between cell cycle control, rates of mutation acquisition, and the loss of functions in tumor suppressor genes [4,5].

Genomic instability and replicative immortality are hallmarks of cancer cells [6–8]. In normal cells, once telomere shortening reaches critical levels, a molecular signal is activated, consequently inducing the state of senescence or apoptosis, allowing protection of genome integrity [9–12]. Short telomeres and high levels of telomerase expression are often reported in human cancers as an intrinsic consequence of tumor genomic instability [13–15].

Telomere critical shortening, and consequently genomic instability, is avoided by the activation of the response to DNA damage. DNA damage causes a halt in cell cycle progression, and there are checkpoint proteins that block this progression to the S phase so that the genetic material is repaired. If the DNA damage is extensive, making it impossible to repair, then the pathways to trigger senescence and death are activated [9,15,16]. The beginning and

progression of the carcinogenesis process is marked by the loss of function of DNA repair genes, which contributes to genomic instability, promoting cancer progression [17–19].

The study of molecular mechanisms and the genetic pathways that lead to tumor genomic instability is a challenge and has been widely studied over the years in several tumor models [3,13]. *WRAP53* is responsible for an antisense transcript of p53 and also encodes a protein with WD40 domains that acts as a scaffold protein participating in important cellular events such as telomerase assembly, the formation of Cajal bodies, and DNA double-strand break repair [20–22].

WD40 repeat motifs (WDRs) range from 40 to 60 amino acids, containing a conserved glycine-histidine motif at the beginning and being terminated with tryptophan dipeptides (W) and aspartic acid (D). Interaction with multiprotein complexes occurs in their existing WD40 repeats within a domain [23–25]. Several WD-repeat proteins are encoded in the human genome and are involved in cellular activities such as chromatin assembly, gene transcription, RNA metabolism, cell cycle regulation, and apoptosis [23–28]. WD-repeat proteins are already known to be involved in tumorigenesis (Receptor for Activated C Kinase 1 (RACK1), cilia and flagella associated protein 52 (WDRPUH), Endonuclein (PWP1) and serine/threonine kinase receptor-associated protein (STRAP)) and also act as tumor suppressants (F-box and WD repeat domain containing 7 (FBW7) and serina/treonina quinase 11 (STK11)) [29–34].

Mutations in WRAP53β are responsible for disorders such as spinal muscular atrophy (SMA), a neurodegenerative disease that in its most common form causes death by age two, and congenital dyskeratosis, a biological disorder associated with telomere shortening. Mutations in its WD40 domain impair telomerase traffic to telomeres, resulting in their progressive shortening. Overexpression of *WRPA53* is linked to carcinogenic transformation, indicating an oncogenic property [35–43].

In this context, it is of great importance that we always strive for innovation in the search of new strategies for cancer management. The identification of new biomarkers that may be efficiently targeted and provide a significant improvement to a patient's prognosis is a crucial step in this search and a current goal in many oncologic studies [44]. In this review, we seek to discuss the cellular roles of *WRAP53*, its possible pathways in carcinogenesis as an oncogene, and a molecular biomarker to be investigated in cancer prognoses.

## 2. WRAP53 Characterization and Cellular Roles

The *WRAP53* gene is found in chromosome 17 and codifies both an antisense transcript for *TP53* stabilization (WRAP53α) and a functional protein containing WD40 repeats that regulates telomere elongation and DNA double-strand break repairs (DDRs), referred to as WRAP53β, WDR79, or TCAB1. The WRAP53γ transcript remains, with its functions unknown [22,45,46].

The protein WRAP53β may be found both in the cytoplasm, where it is responsible for translocation of the survival of motor neuron 1 (SMN1) protein across the cell, and in highly active metabolic regions in the nucleus, known as Cajal bodies [36]. Initially described as a nucleolar accessory body, these structures were originally identified in 1903 by Santiago Ramón y Cajal [47].

Cajal bodies are involved in important nuclear functions such as ribonucleoprotein maturation, RNA polymerase assembly, and telomerase biogenesis [48–51]. They are characterized by the presence of the protein coilin which, due to its interaction with other proteins and RNAs, probably plays a structural role in the assembly of Cajal bodies (Figure 1) [52,53]. Reductions in cellular levels of WRAP53β or its overexpression lead to the rupture of these bodies and prevents the formation of new Cajal bodies, also causing an incorrect location of coilin to occur in the nucleoli [36]. These structures are composed of a diversity of specific ribonucleoproteins (RNPs) that are complexes composed of a non-coding RNA and its associated proteins. This includes small nuclear spliceosomal RNPs (snRNPs), Cajal body specific RNPs (scaRNPs), nucleolar RNPs (snoRNPs), and RNP telomerase components [35,54,55]. Stable RNAs from eukaryotic cells go through extensive post-transcriptional modifications that are much more abundant in ribosomal RNA (rRNA) and small nuclear RNAs (snRNAs). The modifications to rRNAs are carried out in the cell's nucleoli by snoRNPs, while snRNAs are guided to Cajal bodies for further modification by scaRNPs [21,54,56].

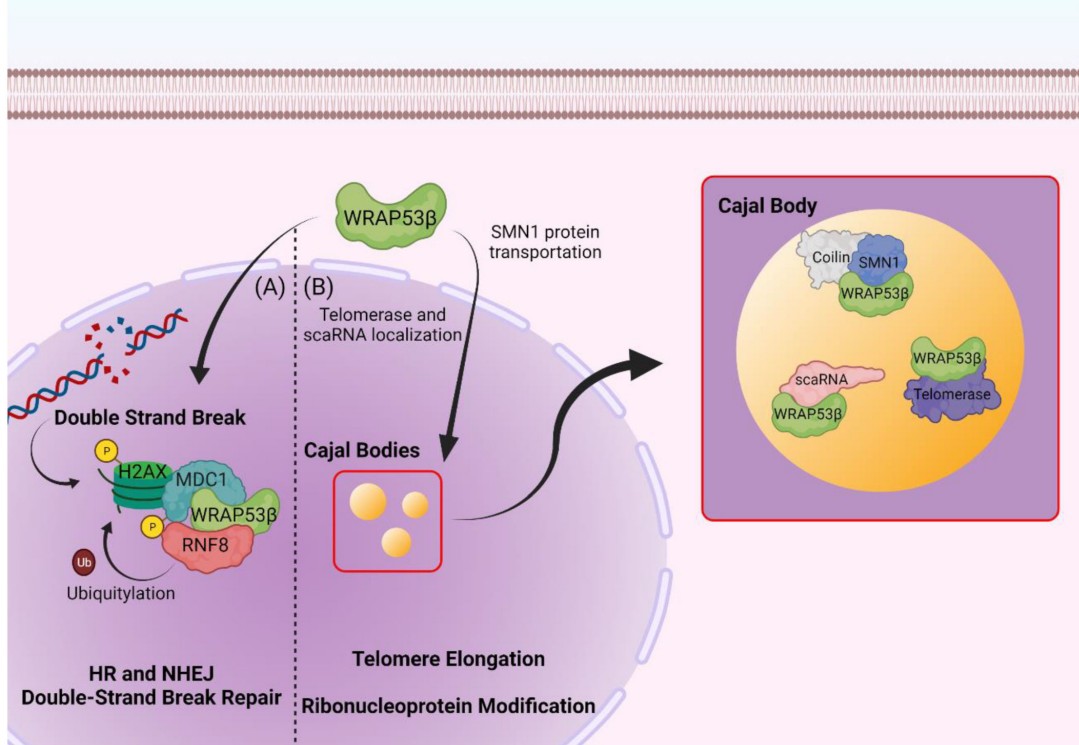

**Figure 1.** WRAP53β roles in cellular homeostasis. (**A**) WRAP53β mediates MDC1 and RNF8 interaction at DNA double-strand breaks. Phosphorylation of histone H2AX at DNA damage sites by DNA protein kinases induces its binding to MDC1, which in turn binds to RNF8 through WRAP53β-mediated activity. RNF8 then ubiquitylates the phosphorylated histone and triggers recruitment and accumulation of DNA damage repair machinery at the break point. (**B**) WRAP53β is essential for Cajal body stability and nuclear function maintenance. WRAP53β mediates SMN1 protein localization in Cajal bodies through transport from the cytoplasm to the nucleus. WRAP53β is also responsible for interacting with and ensuring the activity of the metabolic active telomerase enzyme and for correctly localizing scaRNAs to Cajal bodies, where they will mature and suffer post-transcriptional modifications. Created with BioRender.com (accessed on 30 March 2022).

### 2.1. WRAP53 and Telomerase

WRAP53β associates with scaRNAs as well as telomerase RNA (TERC) and directs them to Cajal bodies for post-transcriptional modifications [21]. C/D box domain scaRNAs are linked to the methylation of snRNAs, while scaRNAs containing H/ACA box domains are responsible for uridine isomerization [20,21,57,58].

TERC or hTR, when used to refer to human telomerase, is an scaRNA of the H/ACA class. The localization of scaRNAs to the Cajal bodies is accomplished through a common element known as the CAB box, where WRAP53β associates directly to promote correct RNA targeting. Mutations in the CAB box that disrupt the interaction of WRAP53β or its depletion result in in a mislocalization of scaRNAs to the nucleoli [20,21,58].

Telomere elongation happens through human telomerase (hTERT) activity, an enzyme that specializes in the synthesis of TTAGGG repeats at the chromosome's ends [59,60]. Mature telomerase enzyme contains TERC alongside a complex of associated proteins and the telomerase reverse transcriptase (TERT) [6,60]. WRAP53β associates to TERC, localizing the telomerase complex to the Cajal bodies and later to the telomeres themselves [20,21].

Due to its binding to telomerase's core components but not to the assembly factors, WRAP53β is considered an important active component of the enzyme, as much as it is important for the proper localization of telomerase in Cajal bodies and its activity in proper DDR, which is essential for genomic stability, and mutations that disrupt these functions may be correlated with cancer progression [20,61,62].

## 2.2. DNA Repair

As a response to DNA double-strand breaks, the cellular machinery has at least five major repair pathways, which are homologous (HR) or non-homologous (NHEJ) repair by nucleotide excision (NER), mismatch repair (MMR), recombination pathway orbase excision repair (BER), and by forming protein complexes that accumulate at sites of damage [17,63,64]. WRAP53β is directly involved in both pathways of double-strand break repair by mediating the interaction of ring finger protein 8 (RNF8) and mediator of DNA damage checkpoint 1 (MDC1) through simultaneously and independently binding to the fork head-associated domains of both proteins [62,65].

The recruitment of ubiquitin-dependent DNA repair factors happens in damage sites where WRAP53β forms a complex with the phosphorylated histone γH2AX alongside MDC1 and RNF8, which is important for RNF8 ubiquitination of proteins in the damaged chromatin and recruitment of the DDR machinery composed of factors such as tumor protein p53 binding protein 1 (TP53BP1), RAD51 recombinase (RAD51), and DNA repair-associated BRCA1 (BRCA1) [17,35,62,65].

The overexpression of WRAP53β allows for a faster repair of double-strand breaks through HR or NHEJ, which points toward the important role of this protein in the orientation of DDR machinery and the maintenance of proper genome integrity [62,66].

## 2.3. Protein Trafficking

Human *SMN1* has been a topic of interest in the health field because despite the wide variety of SMA phenotypes, deletions or intragenic mutations in *SMN1* can be found in all forms of SMA [55,67–69]. The localization and transport of the SMN1 protein to the Cajal bodies is regulated by WRAP53β, and together with gem (Gemin) 2-8 and STRAP, they form the SMN complex responsible for the assembly of snRNPs in the cytoplasm [35,36].

WRAP53β transports the SMN1 protein after cytoplasmic binding by first recruiting it to the nucleus, where it will facilitate interaction with the nuclear pore importinβ and then reach the Cajal bodies [36].

The exact role of SMN1 in Cajal bodies remains unclear, but it is likely involved in more than the transport of newly assembled snRNPs. Depletion of SMN1 disrupts the Cajal bodies, indicating that SMN is vital for the assembly and activity of these structures [70].

## 2.4. WRAP53 and Diseases

Due to its participation in several complex cellular processes, *WRAP53* seems to act both as a tumor suppressor and as an oncogene. Its nuclear or cytoplasmic location may explain some behaviors, being correlated with the regulation of telomerase, survival, and the regulation of factors that involve DNA repair. WRAP53β dysfunction is linked to many diseases which, when associated with the accumulation of DNA damage or defective repair, contribute to the initiation and progression of tumorigenesis [22,35–37,71].

In 1993, *WRAP53* was reported for the first time as a driver for dyskeratosis congenita, a metabolic disturbance associated with telomere shortening and characterized in patients by the triad of dysplastic nails, reticular pigmentation of the upper chest or neck, and oral leukoplakia [37]. This rare hereditary condition encompasses many mutations of the telomerase enzymatic complex, which usually results in bone marrow failure and other manifestations in multiple organs, such as lung and liver fibrosis, developmental defects, and cancer [38–40].

Eleven genes have been reported to promote dyskeratosis: dyskerin pseudouridine synthase 1 (*DKC1*), telomerase reverse transcriptase (*TERT*), telomerase RNA component (*TERC*), TERF1 interacting nuclear factor 2 (*TINF2*), *WRAP53*, NOP10 ribonucleoprotein (*NOP10*), CST telomere replication complex component 1 (*CTC1*), regulator of telomere elongation helicase 1 (*RTEL1*), poly(A)-specific ribonuclease (*PARN*), NHP2 ribonucleoprotein (*NHP2*), and tripeptidyl peptidase 1 (*TPP1*), all associated with telomeric homeostasis [42,43,72–75].

Mutations in WRAP53β associated with dyskeratosis are all found in the highly conserved domain of WD40 repeats. This domain is one of the most important for WRAP53β activity in a variety of cellular processes, serving as a scaffolding for multiple molecule interactions [38,40,42].

Mutations in the WD40 domain result in decreased WRAP53β nuclear levels., which intervene in telomerase traffic to telomeres, and dysfunctional traffic is reported in the most aggressive form of the disease, as it results in the telomeres' progressive shortening [42,43,75].

Telomere shortening may be reverted through hTERT activity. However, this enzyme is strictly limited in human cells [6,76]. Telomerase activity and telomere maintenance are related to cell immortality in cancer, germ cells, and embryonic stem cells [7,8,13,77]. A consequence of disruption or loss of telomere function is chromosome instability, which may lead to cancer progression, genetic fusions, karyotype abnormalities, and predicting poor patient prognoses [78–80]. Leukemogenesis onset, for example, is highly associated with what happens due to loss of structural integrity in the chromosome ends [81].

*WRAP53* implication in leukemias was suggested by Nogueira et al. [82], in which the overexpression of TERT in patients with acute lymphoblastic leukemia (ALL) proved to be a common biomarker indifferent to the ALL subtype. In addition, *WRAP53* showed a strong interaction with TERT in a protein–protein interaction (PPI) network. Dysregulated TERT activity was described as one of the many essential factors for leukemia emergence, and it has been depicted as a common alteration in leukemogenesis [83–85].

Evaluation of telomerase activity, together with the integrity of the telomeres, has emerged as an important prognostic toll in hematological cancers [86–88]. The genomic instability in leukemia derived from telomere disorders is one of the major factors responsible for acquired therapeutic resistance due to karyotype abnormalities and activation of the cell pathways that allow for an escape of the proposed therapeutic mechanisms [89–92].

Another pathological state associated with *WRAP53* mutation-driven pathways is SMA, a disease first described by Werdnig and Hoffmann in 1890 and commonly referred to as the main genetic cause of infant mortality. SMA affects not only motor neurons but also other organs. Respiratory failure is a common cause of SMA in childhood, which in turn can result in death [41,67,93,94].

The reduction in functional SMN1 levels due to defective WRAP53β mediated by defective traffic is seen in severe forms of SMA. The cellular prejudice in this state does not happen due to low availability of SMN1; rather, this is due to defective binding of WRAP53β with SMN1 and the consequent mislocalization of the protein in the cell nucleus [35,36,70].

In addition to progressive telomere shortening, maintenance failures in the Cajal bodies and deficits in WRAP53β-mediated repair of double-strand breaks are also consequences of *WRAP53* mutations (Figure 2). These combined deficiencies may explain, in part, the clinical manifestations seen in patients with dyskeratosis and SMA.

## 3. WRAP53 as a Potential Novel Biomarker for Cancer

*WRAP53* is a natural, highly conserved antisense transcript for the *TP53* gene, a well-known oncodriver, meaning that it can regulate *TP53* expression through mRNA modulation [22, 95–98]. Dependence on a gene for survival or growth of a cancer cell is known as "oncogene dependence" [99,100]. Genetic alterations such as mutations or deletions involving *WRAP53* are present in several types of cancer (Figure 3) (cBioPortal, accessed on 12 August 2022).

A total of 25 articles was included in this review which described the results in patients affected by different types of cancer, such as lung (5), esophageal (1), colorectal (5), hepatocellular (1), ovarian (3), and breast cancer (3), head and neck carcinomas (5), testicular germ cell tumors (1), and nasopharyngeal cancer (1), in addition to studies with cell lines and in vivo models. These data show a *WRAP53* correlation, with important tumorigenesis events highlighted for lung cancer, head and neck tumors, and colorectal carcinomas. We especially focused our discussion on showing the different roles of *WRAP53* in cell biology, showing that structural alterations or alterations in its expression levels can lead to a tumorigenesis process while, on the other hand, its regulation proves to be beneficial in cancer prognosis.

Thus, *WRAP53* overexpression is related not only to carcinogenesis onset but also tumor development and progression, which leads to being pointed out as an oncogene [101]. Table 1 comprises a series of clinical studies that point to malignant onset or the worst patient prognosis due to *WRAP53* mutant expression.

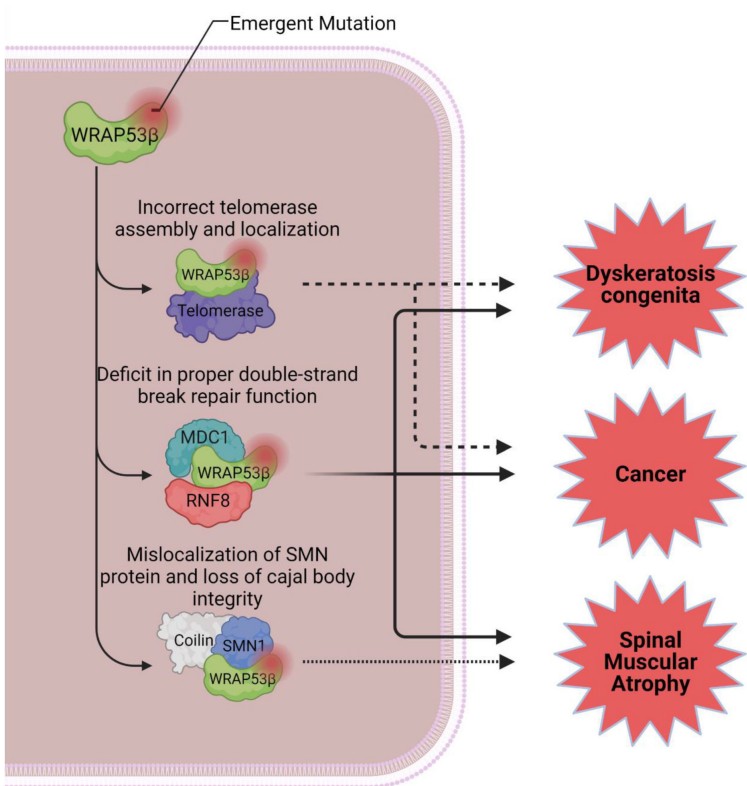

**Figure 2.** Impacts of WRAP53β mutations in pathological states. Improper binding of WRAP53β to telomerase components leads to progressive shortening of telomeres and is highly implied as a determining factor for the severity of dyskeratosis congenita and for the onset of malignancies. Mutations impairing its ability to mediate DNA double-strand break repairs are also worrisome in the general context of genome stability and may be linked to diverse biological events observed in WRAP53β-deficient cells. Lastly, mutations in WRAP53β or in SMN1 that lead to deficient interactions between the two proteins result in mislocalization of SMN1 in the nucleus and consequent loss of Cajal bodies' structural integrity, inducing the spinal muscular atrophy phenotype. Created with BioRender.com (accessed 30 March 2022).

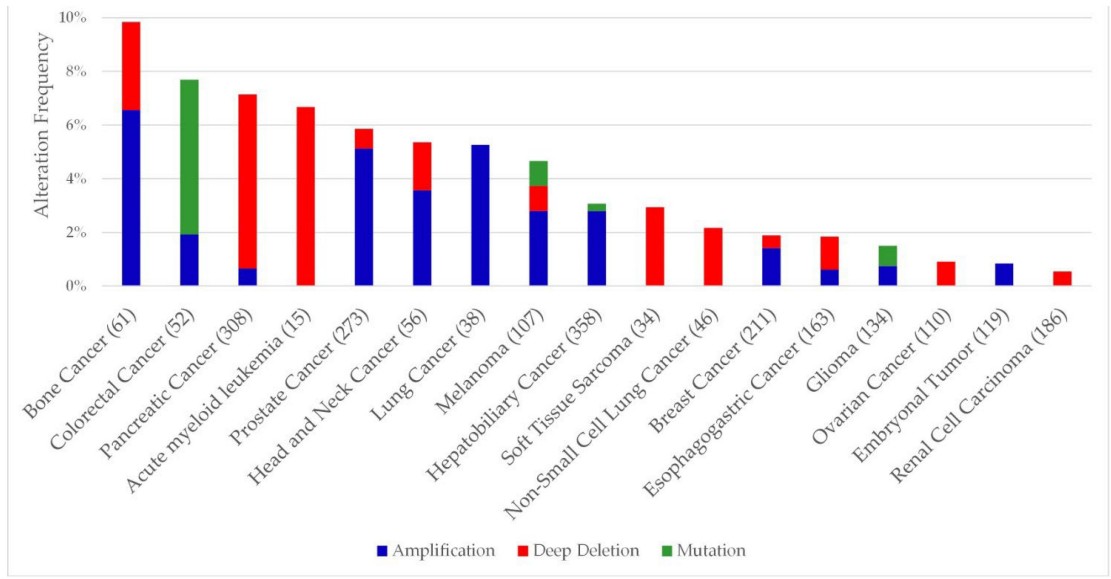

**Figure 3.** Frequency of genetic alterations present in the WRAP53 gene in cancer cell lines. A minimum change limit of 0.5% was applied in the creation of the graph (cBioPortal, accessed on 12 August 2022). The numbers in parentheses represent the number of patients analyzed.

**Table 1.** WRAP53 abnormal expression or mutation in cancer.

| Cancer Subtype | Biological Samples and Methodology | Molecular Mechanisms | Clinical Status | Clinical Outcome or Overall Survival (OS) | Reference |
|---|---|---|---|---|---|
| NSCLC | Patient samples; A549, H1299 95-C, 95-D, HTB182, and HBE and in vivo | WRAP53β plays an important role in tumorigenesis regulating cell cycle progression and apoptosis. | N.I | N.I | [101] |
| | H1299 and A549 cells | WRAP53β induced clonal proliferation through mediation of USP7-MDM2-TP53 pathway. | N.I | N.I | [102] |
| | H1299 and A549 cells | WRAP53β induced clonal proliferation through UHRF1 activity. | N.I | N.I | [103] |
| | Patient samples | WRAP53β overexpression may act as an independent biomarker to predict a poor prognosis. | Patients with primary NSCLC and not previously treated with radiotherapy | WRAP5β expression was an unfavorable prognostic factor related to low overall survival rates. | [104] |
| | Patient samples; A549 and SPC-A-1 cells | Overexpression of WRAP53β reported as a poor prognostic factor, and its downregulation reduced cell proliferation. | Patients with pathological stages I–IV | WRAP53β was present in all patients. | [105] |
| Esophageal squamous cell carcinoma | Patient samples; KYSE150, KYSE180, EC109, and EC9706 cells | Overexpression of WRAP53β was correlated with tumor infiltration, clinical stage, and lymph node metastasis. | Patient's selected at the first diagnosis and without the use of radiotherapy | Presence of altered expression of WRAP53β in 95.6% of cases. | [106] |
| Colorectal cancer | Patient samples | WRAP53β is overexpressed in the primary rectal tumor compared with the normal mucosa. | Patients with stages I, IIA+IIIA+IIIB, and IIIC+IV of the disease | In the group without radiotherapy or metastasis, WRAP53β expression was associated with a worse prognosis. | [107] |
| | in silico analysis of clinicopathological features | WRAP53β is a biomarker of prognoses for young patients. | Patients with stages: I, II, III, IV, and stage 0 and missing cases | WRAP53β was shown to be differentially expressed between the young and elderly groups. | [108] |
| | Patient samples | Overexpression of cytoplasmic or nuclear WRAP53β is indicative of poor prognosis. | N.I | Patients with high expression of cytoplasmic WRAP53β have a low OS and DFS, while its nuclear presence impairs the radiotherapy response. | [109] |

**Table 1.** *Cont.*

| Cancer Subtype | Biological Samples and Methodology | Molecular Mechanisms | Clinical Status | Clinical Outcome or Overall Survival (OS) | Reference |
|---|---|---|---|---|---|
| | Patient samples | Overexpression WRAP53β is reported for colorectal cancer. | Patients with stages I, II, III, and IV of the disease | Cancer patients in the necrotic state had a strong expression of WRAP53β. | [110] |
| | In silico analysis of TCGA database; SW480, HT-29, HCT116, and LoVo cells and in vivo | The elimination of WRAP53β reduced tumor cell proliferation and invasion. | N.I | Higher expression of WRAP53β was observed in colorectal cancer tissues than in normal tissues. | [111] |
| Hepatocellular carcinoma | Patient samples | WRAP53α has clinical value as a promising biomarker with precision in primary screening in hepatocarcinoma and HCV. | Patients with primary HCC and patients with chronic HCV infection | Patients with positive WRAP53α RNA are related to a lower DFS. | [112] |
| Ovarian cancer | Patient samples | The rs2287498 polymorphism is associated with increased risk of invasive ovarian cancer. | Patients with invasive epithelial ovarian cancer in non-Hispanic white women | N.I | [113] |
| | Patient samples; A2780 and SKOV-3 cells | Low nuclear expression of WRAP53β correlates with aggressiveness and poor prognosis of epithelial ovarian cancer. | Patients with serous, endometrioid, mucinous, and other tumors | Aggressive disease, poor prognosis, and reduced survival of the patients. | [114] |
| | Patient samples | SNPs located in *WRAP53-TP53* regions rs1042522, rs2287497, and rs2287498 are more strongly associated with a risk of ovarian cancer. | N.I | N.I | [115] |
| Breast cancer | Patient samples | WRAP53β was shown to be a potential prognostic biomarker. | Patients with a primary breast tumor | The cellular localization of WRAP53β is linked to prognosis and OS. | [116] |
| | Patient samples | Polymorphisms linked to *TP53* or *WRAP53* rs2287499 and rs2287498 may be associated with estrogen receptor (ER)-negative breast cancer. | Patients with stage I and II disease and invasive breast cancer | N.I | [117] |
| | Patient samples and in silico analysis | SNPs rs2287499 and rs1042522 may play an important role in breast cancer susceptibility. | N.I | N.I | [118] |

**Table 1.** *Cont.*

| Cancer Subtype | Biological Samples and Methodology | Molecular Mechanisms | Clinical Status | Clinical Outcome or Overall Survival (OS) | Reference |
|---|---|---|---|---|---|
| Head and neck carcinomas | Patient samples; U2OS, HeLa, H1299, HEK293, HCT116, HDF, AG06814, and MCF10A cells | Overexpression of WRAP53β is a marker of poor prognosis. | N.I | WRAP53β expression levels were higher in patients with recurrent disease. | [119] |
| | Patient samples; HSC-3, Cal-27, and CNE1 ACC2 cells and in vivo | WRAP53β is overexpressed in clinical specimens as well as carcinoma cell lines. | Human nasopharyngeal carcinoma tissue samples and nasopharyngitis tissues | N.I | [120] |
| | Patient samples | Cytoplasmic WRAP53β is a potential predictive marker for poor response to chemoradiotherapy. | Patients treated for primary stage T2N0 or T3N0 glottic laryngeal SCC | A worse DFS and a tendency for worse OS in those patients where WRAP53β was more present in the cytoplasm compared with patients with nuclear staining. | [121] |
| | Patient samples; LK0412 and LK0949 cells | WRAP53β plays an important role in radiotherapy response, and its nuclear localization may be a promising biomarker for overall survival. | Patients with stages T1,T2,T3, N0, N1, and N2 | Patients with nuclear expression of WRAP53β demonstrated greater overall survival than those with non-nuclear staining. | [122] |
| | Hep-2 cells | WRAP53β can be an ideal target for increasing radiosensitivity. | N.I | N.I | [123] |
| Testicular germ cell tumors | In silico analysis of 168 unique telomere-related genes | Overexpression of *WRAP53* can induce telomere lengthening. | Primary tumor samples | In testicular germ cell tumors of the non-seminoma, subtype *WRAP53* was overexpressed. | [124] |
| Nasopharyngeal carcinoma associated with EBV | Patient samples; CNE1, CNE1-LMP1, NP69, HOK, and B95-8 cells | EBV increases the expression of WRAP53β in vitro and, consequently, overactivates the enzymatic activity of telomerase. Its downregulation reduced cell proliferation. | N.I | N.I | [125] |

WRAP53: WD repeat containing antisense to TP53; NSCLC: non-small cell lung cancer; USP7: ubiquitin specific peptidase 7; MDM2: MDM2 proto-oncogene; TP53: tumor protein p53; UHRF1: ubiquitin-like with PHD and ring finger domains 1; SNPs: single-nucleotide polymorphisms; EBV: Epstein–Barr virus; OS: overall survival; N.I: not informed; DFS: disease-free survival; HCC: hepatocellular carcinoma; HCV: Hepatitis C virus.

## 4. WRAP53 as a Prognostic Factor in Cancer

Abnormal cellular molecular pathways triggering uncontrolled cell proliferation is one of the main recent research topics in cancer. Currently, the overexpression of WRAP53 transcripts has been pointed out as a biomarker for a large cohort of cancer subtypes, such as colorectal, hepatocellular, head and neck, breast, ovarian, and esophageal squamous cell cancers [106–108,112,114,116,119,120].

Kamel et al. [112] analyzed WRAP53α expression in conjunction with other prognostic factors and showed that high expression can serve as a novel diagnostic and prognostic biomarker in hepatocellular carcinoma (HCC) and hepatitis C (HCV). In addition, WRAP53α was useful in verifying recurrence-free survival that considers the time from diagnosis to the development of the first evidence of the disease. About 60% of patients who are diagnosed with HCC are at an advanced disease stage with the occurrence of metastases, since early disease detection usually fails due to the absence of specific symptoms. However, patients who are diagnosed at an early stage have a 3-year survival global rate with surgical intervention of >93%, and thus, biomarkers allowing the detection of HCC at an early stage are necessary to improve the prognoses of these patients [126–128].

Overexpression of WRAP53β is observed in a wide range of human cancer cell lines [119]. Recent studies indicate that its overexpression is also related to telomerase activation and the depth of tumor invasion and lymph node metastasis, suggesting a potential role of WRAP53β in cellular mobility and immortality and partly explaining its oncogenic properties, as telomerase reactivation is reported in 90% of all human cancers [106,111,124,129].

Aside from being reported in primary colorectal cancer tumors, overexpression of WRAP53β has also been reported in rectal tumors with ongoing necrosis and is indicated as a poor prognostic factor [107,110]. In colorectal cancer, tumor necrosis is observed in more advanced stages of the disease and is a marker of poor prognosis [130,131].

In non-small cell lung cancer (NSCLC), overexpression of WRAP53β promoted cell proliferation [101–103]. It was observed that WRAP53β localized and interacted with USP7, reducing the ubiquitination of MDM2 proto-oncogene (MDM2) and p53, prolonging its half-life and increasing its stability [102]. It is already known that proteins with WD40 repeats play an important role in the ubiquitin–proteasome processes, allowing the formation of multiple protein complexes [25,132,133]. Until then, this was an unknown function of WRAP53β.

On the other hand, WRAP53β knockdown impairs the growth of cancer cells, inducing cell cycle arrest and apoptosis through mitochondrial pathways. This decreases the potential for invasion both in vitro and in vivo and increases the radiosensitivity of these cells [101,105–107,111,119,120,123], but it has no effect on the radiosensitivity of normal human fibroblasts, which indicates that cancer cells are overly dependent on WRAP53β [119]. In addition, the knockdown of WRAP53β in cancer cells induces massive apoptosis within 48 h to 72 h, reducing tumor growth [101,105,111,119,120,123], while compared with telomerase, silencing this result is expected within 4 weeks [134].

The knockdown effects of WRAP53β on NSCLC are related to negative regulation of cyclins (cyclin D1 and cyclin E), CDKs (CDK2, CDK4, and CDK6) and checkpoint proteins P-RbS795 and S807/811 [101,105]. It is already known that one of the processes for the cell cycle to occur is the activation of protein kinase followed by cyclins and CDKs [135]. Negative regulation of WRAP53β induced a stop in the G1 phase of the cell cycle, which in turn prevented progression to phase S [101,105,111]. Apoptosis mediated by WRAP53β occurs through the activation of caspase-9 and caspase-3 [101], which is consistent with other studies [119,120].

Yuan et al. [105] identified 534 proteins interacting with WRAP53β using A549 cells through bioinformatics tools and identified the following proteins: DKC1, GAR1, RUVBLI, HSPA2, and PKM, which are proteins that function in the processing and modification of the rRNA, components of RNA polymerase II, and delivery of aminoacyl tRNAs and participate in aerobic glycolysis [136–139]. The interaction of WRAP53β with these key

proteins provides new insights into the understanding and new discoveries of mechanisms of interaction.

Furthermore, microarray data of the head and neck carcinomas cellular line shows that WRAP53β might affect multiple processes and cellular pathways, such as the p53 signaling pathway, apoptosis pathway, cell cycle, JAK-STAT signaling pathway, and PI3K-AKT signaling pathway [120]. As previously mentioned, WRAP53β depletion affects cancer cell survival by influencing proliferation and apoptosis. Therefore, its involvement in these pathways was expected. The dysregulation of the pathways associated with these biological processes is related to neoplasms and autoimmune diseases [140–143].

WRAP53β expression is related to the depth of tumor invasion [106]. In vitro studies indicated that WRAP53β knockdown in different cell lines (Cal-27, ACC2, HSC-3, and HCT116) reduced the tumor cell invasion capacity, showing that WRAP53β influences and facilitates tumor cell invasion [111,120]. Invasion is one of the main features present in advanced-stage tumors in several cancer subtypes [144–146].

Zhu et al. [111] evaluated the oncogenic potential of WRAP53β using animal models with colorectal cancer where, after removal of the xenografts, it was observed that animals with knockout of WRAP53β expression showed a decrease in the formation and growth of tumor cells. The in vivo findings are consistent with the studies of Sun et al. [120], who also evaluated this potential using xenografts in mice with oral squamous cell carcinoma Cal-27 cells. The analysis revealed that both the volume and weight of the tumors were significantly lower in the WRAP53β knockdown mice when compared with the control group.

The first time *WRAP53* was described in cancer was after an analysis of the common genetic variation of *TP53* and its flanking genes, and these data showed that the single nucleotide polymorphisms (SNPs) rs2287499 and rs2287498 were significantly linked to increased development of ER-negative breast cancer [117], and the same rs2287498 SNPs have also been linked to aggressive ovarian cancer [113,115]. The presence of SNPs in the *WRAP53* gene generates an amino acid change, namely rs2287499 (R68G), an Arg/Gly polymorphism, and rs2287498 (F150F), a Phe/Phe polymorphism [113,117,118], and the presence of these alterations in *TP53* and *WRAP53* may affect their products, causing a vulnerability to cancer and failure to respond to therapy [115,147]. Epidemiological studies have extensively discussed the involvement of polymorphisms in a variety of cancers [148–150].

To analyze the potential of WRAP53β in selecting suitable patients for radiotherapy, Qiu et al. [123] analyzed the effects of its knockdown and associated it with an effective radiotherapy response. Since primary tumors have an increased expression of WRAP53β, radiotherapy may not be effective [107]. Its subcellular location is also taken into account in the evaluation of patients who benefit or do not benefit from radiotherapy. The nuclear localization of WRAP53β is associated with a better response to radiotherapy and a better course of the disease [116,122], while the predominant cytoplasmic location may be a predictive marker of poor response and lower disease-free survival rate [109,121]. Since 60% of tumor patients receive irradiation as part of their treatment [151], a biomarker to select suitable patients for radiotherapy is of extreme relevance.

Hedström et al. [114] demonstrated that low nuclear expression of WRAP53β is correlated with a fourfold increased risk of death from ovarian cancer. Low levels of nuclear expression showed a deregulation of the factors involved in the response to DNA damage, which in turn resulted in an increase in genomic instability. This is consistent with studies that have shown that the loss of the WRAP53β protein results in defective repair of DNA double-strand breaks [62].

Wang et al. [125] reported that the EBV increased the expression of WRAP53β in vitro, which is also likely associated with the overactivation of telomerase, and the involvement of WRAP53β in DNA damage repair pathways may partly account for its overexpression in EBV infections associated with nasopharyngeal carcinoma. One of the mechanisms by which EBV induces cellular immortality is the overactivation of telomerase in both

epithelial cells and B lymphocytes through its main oncoprotein LMP-1 [152–155]. In particular, LMP-1 has been reported to activate TERT via the PI3K-AKT pathway [156].

In short, high levels of *WRAP53* transcripts were observed to be present in a variety of tumor cell lines. In addition, WRAP53β knockdown is associated with in vivo tumor reduction and the induction of apoptosis of tumor cell lines. Human tumor cells are sensitive to WRAP53β inhibition. These findings highlight the role of *WRAP53* in cancer, both in vitro and in vivo, and present it as a potentially promising new target for cancer therapy.

## 5. Conclusions

The clinical and experimental data analyzed in this review demonstrate that excessive telomere shortening, accented telomerase activity, and WRAP53β overexpression are present in a diverse subset of malignant phenotypes. Its different locations and functions render this protein as involved in several important cellular processes, such as the organization and formation of Cajal bodies, repair of DNA double-strand breaks, and the interaction and localization of hTERT and *TP53*. In addition to being commonly reported in cancer, loss of WRAP53β has been associated with dyskeratosis congenita and SMA. However, more clinical and experimental investigations are still needed for a better understanding of the role of *WRAP53* and its transcripts in the mechanisms involved in tumorigenesis so that it may be addressed as potential new biomarker in cancer and a target in the development of new treatments.

**Author Contributions:** Invitation received, C.A.M.-N.; conceptualization, R.B.G., C.B.M. and C.A.M.-N.; provision of data and subsequent analysis and interpretation, R.B.G., C.B.M., L.d.C.P., F.M.C.d.P.P., I.V.B., L.d.C.P., R.M.R., M.O.d.M.F., M.E.A.d.M., A.S.K. and C.A.M.-N.; writing—original draft preparation, R.B.G., C.B.M. and C.A.M.-N.; writing—review and editing, R.B.G., C.B.M. and C.A.M.-N.; funding acquisition, A.S.K. and C.A.M.-N. All authors have read and agreed to the published version of the manuscript.

**Funding:** This study was supported by the Brazilian funding agencies Coordination for the Improvement of Higher Education Personnel (CAPES) to C.B.M., the National Council of Technological and Scientific Development (CNPq grant number 404213/2021-9 to C.A.M.-N. and Productivity in Research PQ scholarships to M.O.d.M.F., M.E.A.d.M., and A.S.K.), and the Cearense Foundation of Scientific and Technological Support (FUNCAP grant number P20-0171-00078.01.00/20 to F.M.C.d.P.P., M.O.d.M.F., and C.A.M.-N.). We also thank PROPESP/UFPA for the publication payment.

**Conflicts of Interest:** The authors declare no conflict of interest. The funders had no role in the design of the study; in the collection, analyses, or data interpretation; in the writing of the manuscript, or in the decision to publish the results.

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
