# Peer review of "The Role of WRAP53 in Cell Homeostasis and Carcinogenesis Onset"

_cimb, doi:10.3390/cimb44110372_

Round 1
Reviewer 1 Report
The interesting topic presents practical information for the scientific/academic community. The reading is fluid.
I recommend that the authors check for plagiarism.
I recommend the authors align the topic with the special issue demonstrating the application of omics to early disease detection.
When using abbreviations for the first time, the complete name should appear.
The title may be like this: THE ROLE OF WRAP53 IN CELL HOMEOSTASIS AND CARCINOGENESIS ONSET
Pay attention to gene nomenclature: https://www.ncbi.nlm.nih.gov/pmc/articles/PMC7494048/
I would like to recommend adding a section with future perspectives with a concise outline of the state-of-the-art studies using nanoparticles to address the underlying pathologies to circumvent some limitations of the current therapeutic applications addressing alterations on the gene profile gene.
Take a look at the following manuscripts:
https://link.springer.com/article/10.1007/s00018-021-03783-0
https://www.mdpi.com/2072-6694/13/3/479/htm
I would like to recommend to the authors to update your list of references.
Author Response
high-quality review and then we present the answers to the questions.
We inform that with the reviews and suggestions, we were able to improve the idea presented by our work and we appreciate the opportunity. We hope this review has left the article suitable for publication in this high-impact journal and respect in the area.
Kind Regards.
Response to reviewer 1
The interesting topic presents practical information for the scientific/academic community. The reading is fluid.
I recommend that the authors check for plagiarism.
R: The review was submitted to plagiarism checking programs as recommended and the necessary corrections were made.
I recommend the authors align the topic with the special issue demonstrating the application of omics to early disease detection.
R: Information on the data available in the TCGA database on the frequency of mutational changes related to the WRAP53 gene in cancer was added to the text.
When using abbreviations for the first time, the complete name should appear.
R: A text revision has been made to correct the issue.
The title may be like this: THE ROLE OF WRAP53 IN CELL HOMEOSTASIS AND CARCINOGENESIS ONSET
R: Title has been corrected as suggested
Pay attention to gene nomenclature: https://www.ncbi.nlm.nih.gov/pmc/articles/PMC7494048/
R: The nomenclature has been revised and corrected.
I would like to recommend adding a section with future perspectives with a concise outline of the state-of-the-art studies using nanoparticles to address the underlying pathologies to circumvent some limitations of the current therapeutic applications addressing alterations on the gene profile gene.
Take a look at the following manuscripts:
https://link.springer.com/article/10.1007/s00018-021-03783-0
https://www.mdpi.com/2072-6694/13/3/479/htm
R: Dear reviewer, we really appreciate your suggestions, however this is not the central idea of our manuscript. Our focus in this article was to present the involvement of the wrap53 gene in carcinogenesis and its biological outcomes. We will consider your suggestion for the next article that will investigate the involvement of the gene in patients with leukemia in our population in Brazil.
I would like to recommend to the authors to update your list of references.
R: We re-evaluated our references to try to reduce the number of review studies to focus on original reports

Reviewer 2 Report
Manuscript is interesting. I have no major criticisms.
Author Response
We inform that with the reviews and suggestions, we were able to improve the idea presented by our work and we appreciate the opportunity. We hope this review has left the article suitable for publication in this high-impact journal and respect in the area.
Kind Regards.
Response to reviewer 2
Manuscript is interesting. I have no major criticisms.
R= Dear reviewer, my co-authors and I would like to thank you for this high-quality review

Reviewer 3 Report
This review is poorly organized and unfocused. It excessively references other reviews instead of original reports. Table 1 is poorly designed as it does not deal with clinical studies (only clinical specimens) or molecular mechanisms (only phenotypes). Very little information is provided about WRAP53alpha - a more focused, in-depth review dealing solely with WRAPbeta would be preferable.
Author Response
Dear reviewer, my co-authors and I would like to thank you for the suggestions made during this high-quality review and then we present the answers to the questions.
We inform that with the reviews and suggestions, we were able to improve the idea presented by our work and we appreciate the opportunity. We hope this review has left the article suitable for publication in this high-impact journal and respect in the area.
Kind Regards.
Response to reviewer 3
This review is poorly organized and unfocused. It excessively references other reviews instead of original reports. Table 1 is poorly designed as it does not deal with clinical studies (only clinical specimens) or molecular mechanisms (only phenotypes). Very little information is provided about WRAP53alpha - a more focused, in-depth review dealing solely with WRAPbeta would be preferable.
R: Dear reviewer, we re-evaluated our references to try to reduce the number of review studies to focus on original reports.
Table I was redesigned to provide more clinical information, but most studies provided little or no information about the clinical status of the patients covered in the study.
As for WRAP53alpha, we chose not to approach it in depth in this review because there is still no data in the literature that explores its function in depth, unlike WRAP53beta, which has been widely explored in recent years.

Reviewer 4 Report
In this manuscript entitled “WRAP53 ROLE IN CELL HOMEOSTASIS AND ASSOCIATIONS WITH CARCINOGENESIS ONSET” Gadelha and colleagues provide a comprehensive review of the literature for the role of WRAP53 in genome integrity. Despite that WRAP53 is poorly investigated, the manuscript provides a clear picture of WRAP53 functions. Overall, the manuscript is well structured and the art very informative.
Comments:
1. Authors should analyze the profile of identified mutations in WRAP53 based on TCGA data, the frequency of these mutations among different types of cancers and report any studies that have characterized WRAP53 mutants.
2. The conclusion section is not well written as there are many syntax errors. I propose authors to carefully revise this section.
3. Figure 1 legend (WRAP53β roles in cell metabolism): WRAP53β roles in cellular homeostasis describes the figure better.
Author Response
Dear reviewer, my co-authors and I would like to thank you for the suggestions made during this high-quality review and then we present the answers to the questions.
We inform that with the reviews and suggestions, we were able to improve the idea presented by our work and we appreciate the opportunity. We hope this review has left the article suitable for publication in this high-impact journal and respect in the area.
Kind Regards.
Response to reviewer 4
In this manuscript entitled “WRAP53 ROLE IN CELL HOMEOSTASIS AND ASSOCIATIONS WITH CARCINOGENESIS ONSET” Gadelha and colleagues provide a comprehensive review of the literature for the role of WRAP53 in genome integrity. Despite that WRAP53 is poorly investigated, the manuscript provides a clear picture of WRAP53 functions. Overall, the manuscript is well structured and the art very informative.
Comments:
- Authors should analyze the profile of identified mutations in WRAP53 based on TCGA data, the frequency of these mutations among different types of cancers and report any studies that have characterized WRAP53 mutants.
R: Dear reviewer, we analyzed the TCGA database to identify mutations in different types of cancer, as requested, but we did not find any studies that characterized WRAP53 mutants.
- The conclusion section is not well written as there are many syntax errors. I propose authors to carefully revise this section.
R: The conclusion has been rewritten to more clearly cover the theme addressed in the review.
- Figure 1 legend (WRAP53β roles in cell metabolism): WRAP53β roles in cellular homeostasis describes the figure better.
R: Figure 1 legend has been rewritten for better understanding.
